# SmS/EuS/SmS Tri-Layer Thin Films: The Role of Diffusion in the Pressure Triggered Semiconductor-Metal Transition

**DOI:** 10.3390/nano9111513

**Published:** 2019-10-24

**Authors:** Andreas Sousanis, Dirk Poelman, Philippe F. Smet

**Affiliations:** Lumilab, Department of Solid State Sciences, Ghent University, Krijgslaan 281/S1, 9000 Gent, Belgium; andreas.sousanis@ugent.be (A.S.); dirk.poelman@ugent.be (D.P.)

**Keywords:** SmS, EuS, semiconductor-metal transition, structural properties, piezoresistivity, interdiffusion, rare earths, thin films

## Abstract

While SmS thin films show an irreversible semiconductor-metal transition upon application of pressure, the switching characteristics can be modified by alloying with other elements, such as europium. This manuscript reports on the resistance response of tri-layer SmS/EuS/SmS thin films upon applying pressure and on the correlation between the resistance response and the interdiffusion between the layers. SmS thin films were deposited by e-beam sublimation of Sm in an H_2_S atmosphere, while EuS was directly sublimated by e-beam from EuS. Structural properties of the separate thin films were first studied before the deposition of the final nanocomposite tri-layer system. Piezoresistance measurements demonstrated two sharp resistance drops. The first drop, at lower pressure, corresponds to the switching characteristic of SmS. The second drop, at higher pressure, is attributed to EuS, partially mixed with SmS. This behavior provides either a well-defined three or two states system, depending on the degree of mixing. Depth profiling using x-ray photoelectron spectroscopy (XPS) revealed partial diffusion between the compounds upon deposition at a substrate temperature of 400 °C. Thinner tri-layer systems were also deposited to provide more interdiffusion. A higher EuS concentration led to a continuous transition as a function of pressure. This study shows that EuS-modified SmS thin films are possible systems for piezo-electronic devices, such as memory devices, RF (radio frequency) switches and piezoresistive sensors.

## 1. Introduction

Materials science plays a significant role in the fabrication of new devices based on chemical compounds with specific properties, providing unprecedented device capabilities. Such a device is the piezoelectronic transistor (PET) [1,2,3,4], which can exploit the pressure-induced semiconductor to metal transition (SMT) of certain compounds. Examples are Sm chalcogenides [5,6], Mott insulators, as well as other material oxides (e.g., Sr_2_IrO_4_) [7]. Specifically, SmS features a hysteretic pressure-induced SMT [6,8,9] at around 0.65 GPa, where the resistance drops significantly. In single crystals, the system returns to the semiconducting state upon release of the pressure, while for thin films thermal annealing or a tensile force is needed. Although this behavior was mainly studied in bulk crystals [6,10], we recently managed to observe hysteretic resistance loops in SmS thin films [11]. Another approach to provide a tunable, hysteretic piezoresistive behavior could be the use of alloyed systems [12,13] that shift the energy bands of the primary material (SmS), without inducing the metallic state, leading to reversible switching characteristics upon releasing force. In order to achieve this, SmS can be alloyed with similar materials that have a somewhat wider band gap. This slightly opens the band gap and shifts the SMT to higher pressures, in comparison to pure SmS. In Figure 1, we see a qualitative representation of the hysteretic discontinuous resistance change in SmS as a function of pressure, as observed in single crystals. This drop can be explained as follows. At atmospheric pressure, the material possesses its semiconducting high resistance state (HRS), which changes to a metallic state (low resistance state, or LRS) when pressure is applied. The required pressure strongly depends on the gap between the *4f* states of Sm ions and the *5d(t_2g_)* degenerate conduction band (CB) [14]. SmS shows a smaller gap (~0.15 eV), than SmSe (transition pressure = 2 GPa) and SmTe (transition pressure = 4.5 GPa), resulting in an isostructural transition at lower pressures (at around 0.65 GPa) [15]. The substitution of SmS with, for instance, wide band gap rare earth-based materials (e.g., EuS or YbS) [16,17], promotes a shift of the *5d* band of the alloyed system to higher energies, with the increase of the substituent further increasing the Sm *4f-5d* band gap. Nevertheless, there are other elements substituting for Sm, which directly promote a chemically triggered transition to the metallic state at ambient conditions. Such elements, for example, are Gd and Y [12,13,17,18], both decreasing the phase transition to even lower pressure. This tunability of the piezoresistive response promotes SmS and alloyed SmS as possible candidates for memory and RF (radio frequency) switching devices [1,2,3,4,19].

In this work, we report on the piezoresistance response of a SmS/EuS/SmS tri-layer thin film system. This tri-layer is an experimentally easy and controlled way to study the alloyed system Sm_1-x_Eu_x_S. Since the individual films are very thin, interdiffusion between the layers is expected, especially at elevated temperatures during deposition or after post-deposition thermal annealing. We first present some of the basic properties of the separate compounds before showing the structural properties of the tri-layer system. We used two different substrate temperatures to study the diffusion between the layers upon deposition. Thermal post-annealing was performed to further investigate diffusion. The resistance drop, and its subsequent rise after pressure release, confirms that this material system is a promising candidate for several strain-based sensing devices. The resistance response strongly depends on the mixing between the layers. The as-deposited tri-layers at 250 °C showed a three states system behavior, while deposition at 400 °C can lead to a conventional system with two states.

## 2. Materials and Methods 

An e-beam evaporator (model: Leybold Univex 450, Leybold GmbH, Germany) was used to deposit SmS on Corning (1737F) glass and 6 inch Si (100) wafers. Sm metal (Smart Elements GmbH, Vienna, Austria, 99.99%) was used as the target material and deposited at a rate of typically 0.8 nm/s under a reactive H_2_S (Praxair Inc., Danbury, CT, USA, 99.8%) flow, leading to a pressure of 1 × 10^−5^ mbar during the deposition. The base pressure of the system was approximately 2 × 10^−6^ mbar. Details on the deposition conditions required for obtaining stoichiometric and well-crystallized SmS are described elsewhere [20]. EuS was first synthesized starting from Eu_2_O_3_ (Alfa Aesar, Thermo Fisher GmbH, Germany, 99.99%) powder. Eu_2_O_3_ was placed in a tube furnace for 2 h under H_2_S flow, at 1000 °C. The produced sulfurized powder consisted of EuS, as confirmed by XRD (not shown). Then, by using a hydraulic press, EuS pellets were prepared as target material for the deposition of EuS thin films by e-beam evaporation under identical H_2_S flow as for the deposition of SmS. Resistivity values were calculated from the sheet resistance measured using a four probe setup. In order to electrically insulate the thin films from the Si substrate, dedicated samples were deposited on top of a 600 nm Al_2_O_3_ (Alfa Aesar, Thermo Fisher GmbH, Germany, 99.99%) thin film, prepared by e-beam evaporation. For electrical measurements where the resistivity was measured across the thin film, iridium bottom and top electrodes (thickness 50 nm) were deposited by e-beam evaporation, starting from an Ir slug (Aldrich Chemistry BVBA, Belgium, 99.9%), with the substrate at room temperature. The bottom electrodes were blanket layers deposited over the entire substrate, whereas the top electrodes were deposited through a mask, yielding circular electrodes with a diameter of 1.5 mm. The SMT was introduced in SmS by rubbing the thin film surface with a round-shaped metal tip, without visibly damaging the thin film surface. The reverse MST (metal to semiconductor transition) was induced by thermal annealing at 400 °C in vacuum. The semiconducting and metallic SmS state will be referred to as S-SmS/HRS (high resistance state) and M-SmS/LRS (low resistance state), respectively.

Structural characterization of the fabricated thin films was carried out via X-ray diffraction (XRD) using a standard powder diffractometer (D8 with Ni-filtered CuKα_1_ radiation, λ = 0.154059 nm, Bruker AXS GmbH, Karlsruhe, Germany). In situ high temperature X-ray diffraction (HTXRD) patterns were also measured using a Bruker D8 Discover system (equally using CuKα_1_ radiation) with an integrated annealing chamber, able to support several atmospheric conditions. In the latter case, a linear detector was used, which allowed for collecting diffraction patterns in seconds, without moving the sample or detector. SEM analysis was carried out in an FEI electron microscope (Quanta FEG 200, Hillsboro, Oregon, USA), with a point resolution of 1.7 nm at 20 kV. Ultraviolet-visible (UV-Vis) spectra were recorded at room temperature in the specular reflectance geometry (V-W method) with a Varian Cary 500 UV-Vis spectrophotometer (Agilent, Santa Clara, CA, USA) in the wavelength range of 200–800 nm. Piezoresistance measurements were performed using a homemade device, similar to reported devices [21]. Finally, we used X-ray photoelectron spectroscopy (XPS) to detect and confirm the presence and diffusion of the components in the multi-layered structures. The used set-up was an ESCA S-probe VG (Thermo Fisher GmbH, Germany) with an Al(K_a_) source (1486.6 eV). The base pressure of the system was 5 × 10^−10^ mbar, while the pressure during Ar-sputtering increased up to 2 × 10^−7^ mbar. The sputter time was 25 s per step and the measurement time after each step was 5 min, unless mentioned otherwise. In all studies of the tri-layers, the Sm *3d_5/2_*, S *2p*, O *1s*, and Eu *3d_5/2_* peaks were used for the calculation of elemental concentrations. 

## 3. Results

### 3.1. Individual SmS and EuS Thin Films

EuS thin films with a thickness of 25 nm are largely transparent in the visible region, although the UV-Vis spectra show a broad absorption band between 1.8 and 2.8 eV, caused by the transition from the *4f^7^* ground state to the *4f^6^5d* configuration in Eu^2+^ (Figure 2a). This is in line with EuS being a natural ferromagnetic semiconductor [22], showing an optical band gap of 1.65 eV for thin film EuS [23]. EuS thin films with a thickness of 25 nm deposited at 250 °C on Si (100) wafer show a good crystallinity with preferential growth orientation of the (200) planes parallel to the substrate. Notice that SmS and EuS have the same rock salt lattice structure, with almost identical lattice constants for S-SmS and EuS (5.970 Å and 5.968 Å respectively). Consequently, standard XRD analysis cannot be used to discriminate between both materials. 

Applying pressure to these EuS thin films does not introduce any transition. This is different for the SmS thin films, where moderate pressure can provide the SMT at room temperature. Usually, soft polishing is used to induce M-SmS [24,25]. Here, gently rubbing the sample surface leads to the SMT, with the accompanying reduction in lattice constant. This is demonstrated by the corresponding XRD peaks shifting to higher 2θ values (Figure 2b), in accordance with previous investigations [26]. The 100 nm S-SmS thin films show resistivity values in the range from 1.5 to 5 × 10^−1^ Ωcm (as derived from the sheet resistance), while the rubbed 100 nm M-SmS layers demonstrated 3–4 × 10^−3^ Ωcm. These values are comparable to previous investigations [27]. The as-deposited EuS thin films showed a sheet resistance of about 1.7 MΩ (25 nm EuS thin film), which is almost one order of magnitude lower, relative to previous investigations on highly optimized insulating EuS thin films (the sheet resistance is higher than 20 MΩ for thicknesses between 20 and 200 nm) [28]. 

In the SmS/EuS/SmS tri-layers, we studied the diffusion process in this nanocomposite system in order to tune the piezoresistive behavior. The substrate temperature, post-deposition annealing, as well as the thickness of the diffusive layer (EuS) relative to the full tri-layer thickness were changed to influence the layer interdiffusion.

### 3.2. Deposition and Annealing of Tri-Layers

Figure 3 shows an XRD plot of a 35/30/35 nm SmS/EuS/SmS tri-layer deposited at 250 °C, before (black curve) and after rubbing (red curve). After rubbing the sample surface, we observe two peaks for the (200) lattice plane, one related to EuS and one to M-SmS. Hence, at a deposition temperature of 250 °C, the individual layers do not thoroughly mix. XPS analysis, discussed below, confirms that the thin films are not mixed when deposited at 250 °C.

In order to probe the influence of temperature on the structural evolution of the tri-layer thin films, in situ HTXRD was performed up to 800 °C (with a heating rate of 10 °C/s). Figure 4a shows the in situ HTXRD patterns of an as-deposited (at 250 °C) 35/30/35 nm SmS/EuS/SmS tri-layer under 20% O_2_ and 80% He atmosphere. In this atmosphere, the tri-layer system remains stable up to 350 °C. As mentioned before, the peak at around 30° is composed of the reflection from the (200) lattice planes of SmS and EuS. Above 350 °C, the diffraction intensities decrease—although the (111) reflection of SmS and EuS remains visible—and at 500 °C the observed diffraction positions match with those of Sm_2_O_2_S. For comparison, Figure 4b shows the stable character of 25 nm EuS thin film up to 450 °C in the ambient-like atmosphere, in line with previous investigations [28,29].

Then, we increased the substrate temperature to 400 °C to explore any changes in the switching properties and the degree of mixing in the tri-layer thin film stacks. In order to study the switching behavior, we rubbed the surface of a triple layer to induce the SMT, as shown earlier for the single SmS film (Figure 3). A clear SMT was still observed by following the shift of the (200) the diffraction peak (Figure 5a) from about 30.27° to 31.44° (corresponding to a change in d_200_ from 2.95 Å to 2.84 Å), although a smaller peak remained at the original position. While the former peak demonstrated a behavior typical for SmS, the latter peak is indicative of the presence of an EuS-like part in the tri-layer, which does not show switching. A similar picture is observed for the (111) peak. By annealing in vacuum at 400 °C, the semiconducting state can successfully be recovered (Figure 5a, blue curved), although the derived d_200_ lattice spacing of 2.92 Å is somewhat smaller than the original value. 

Subsequently, we investigated the thermally induced metal-semiconductor reverse transition (in ambient atmosphere) of the tri-layer system deposited at 400 °C. The red curve on the bottom of Figure 5b corresponds to the rubbed (without any annealing) tri-layer initially deposited at 400 °C. This sample was fabricated following the same process as in Figure 5a for the red curve. Then, 10 min of post-annealing at 300 °C in air was performed, which did not yield any significant structural change. Prolonging the accumulated annealing time to 25 or 60 min did not switch back the M-SmS part of the thin film. The back switching of the tri-layer to the semiconducting state (smaller 2θ value) was observed, however, after only 10 min of annealing at a slightly higher temperature of 350 °C. In contrast, a single SmS film is quite stable, without showing any back switching behavior, when it is annealed at 350 °C, for 30 min in air (purple dashed curve in Figure 5b).

### 3.3. Piezoresistive Behavior

Figure 6a shows the piezoresistance of the tri-layer system of SmS/EuS/SmS, deposited at 250 °C, with layer thicknesses of 35/30/35 nm, using Ir bottom and top electrodes. It is important to notice that the application of pressure with the indenter leads to a relatively small area (of the order of tens of μm^2^) where the pressure is applied, as compared to the total top electrode area (1.8 mm^2^). Hence, the measured resistance across the thin film is essentially determined by two parallel resistors, one with variable resistance (where the pressure is applied) and one with fixed resistance (outside the indenter area). The values given below are for the combined resistance, as it is difficult to estimate the pressed area, given that it is a function of the applied force and thus the contact area of the indenter. When the indenter just makes contact with the electrode, the resistance is 76 kΩ (HRS). Application of pressure first leads to a limited, gradual decrease in the resistance, until a first sudden drop to 5.2 kΩ appears at a force of about 0.5 N. We define this as an intermediate resistance state (IRS). Note that the actual change in resistance from the semiconducting to the metallic state is higher, taking into account the effect of the parallel resistance. The force threshold is similar to the behavior of single SmS thin films, leading to the conclusion that the SmS-like parts in the tri-layer switch first. Looking at higher force values, we see a second sharp resistance drop at around 1.55 N (to a resistance of 260 Ω), which is likely related to a partly mixed alloy of SmS and EuS. The resistance gradually drops further upon increasing force, to 160 Ω at 2 N, which can be related to the piezoresistive behavior of Eu-rich SmS, where the pressure leads to a narrowing of the *4f*–*5d* gap [16]. Upon gradual release of the pressure, the resistance increases slowly again, showing a major, rather discontinuous, increase between 0.8 and 0.5 N. This is likely related to the switching back of the alloyed part of the tri-layer, which had switched to an LRS around 1.5 N. Just before full release of the force (when the contact between the indenter and the top electrode is lost), the resistance is still an order of magnitude below the initial resistance of 76 kΩ. When contact is made again with the top electrode, the high resistance value is restored, indicating that the switching back of the relatively pure SmS part of the tri-layer (which had switched around 0.5 N during the loading) occurs very close to ambient pressure. As demonstrated in the inset of Figure 6a, the same piezoresistive behavior is found during 5 consecutive cycles of loading and unloading, showing that effectively a three state system, between an HRS, IRS and LRS, is obtained. It should be mentioned that the pressure needed for the second drop slightly changes upon cycling, with a variation between 1.3 and 1.6 N. A more integrated measurement approach where pressure is applied more uniformly is currently under development in order to further characterize the piezoresistive behavior.

In Figure 6c, we demonstrate the piezoresistance of a sample with identical composition and electrode configuration, deposited at a substrate temperature of 400 °C. A significant difference with the previous case (sample deposited at 250 °C, Figure 6a) is that no sudden resistance drops are observed, at least not with large changes in the resistance. As will be demonstrated below by means of XPS depth profiling, the tri-layers deposited at 400 °C show a higher degree of mixing between the individual SmS and EuS layers. The high resistance state shows values of about 29 kΩ, smaller than the corresponding value (76 kΩ) for the tri-layer deposited at 250 °C, while at higher force values (~1.6 N) the resistance reaches a value of 421 Ω.

A pure EuS layer does not show significant changes in the resistance up to pressures similar to those where the SMT appears in single SmS films. This is due to a larger gap between the *4f* and *5d* bands for EuS compared to SmS. A structural change does occur, from fcc to bcc, at around 20 GPa [30], while 36 GPa is needed for the complete structural change, accompanied by a change in the valence state [31]. For a 100 nm EuS layer in our experiment, with bottom and top electrodes, the resistance across the EuS thin film was 2.2 × 10^7^ Ω at close to ambient pressure (when the indenter just made contact with the top electrode), while it remained at a high value of 1 × 10^7^ Ω, at 1.5 N. Measuring across the 100 nm EuS layer, the observed values were similar to those recorded for high quality EuS layers, via sheet resistance measurements, as mentioned above. The value of 2.2 × 10^7^ Ω is slightly higher than expected from the sheet resistance for 25 nm EuS.

### 3.4. XPS Analysis

To correlate the piezoresistance properties of the studied tri-layer system to the extent of inter-diffusion between the three individual layers, XPS depth profiles were recorded for tri-layers with different compositions and annealing conditions. Photoelectrons from the four main elements (Sm, S, O, Eu) were recorded at the distinct spectral regions for each component: Sm *3d_5/2_*, S *2p*, O *1s*, and Eu *3d_5/2_*. The presented results cover the total thickness of the tri-layer system, while the Si substrate is not included.

An as-deposited composite thin film, at 250 °C, does not yield a fully alloyed system (see Figure 7a). The Sm concentration decreases from approximately 60% in the outer parts of the stack to 24% in the mixed region, where the Eu concentration reaches 40%, while the S concentration remains relatively constant throughout the sample. Care must be taken in the interpretation of these XPS results, since the concentration distribution can be influenced by knock-on sputtering, obscuring the interfaces between films. In addition, it was found by the authors of [32] that any exposed SmS surface oxidizes, even in UHV (ultrahigh vacuum) conditions. Nevertheless, the oxidation is surface limited, with the O remaining at very low concentration in the main volume of the tri-layer, even after 4 h of annealing at 400 °C (blue curve in Figure 7c). However, the main conclusion from Figure 7a remains, namely that there is incomplete mixing of the layers after deposition at a substrate temperature of 250 °C.

In this case, the tri-layer system is mainly consisting of the SmS-like parts that show a pressure-triggered resistance and color change at low applied pressure, and the partly mixed EuS/SmS intermediate thin film. Before the application of pressure, both materials (SmS, EuS) demonstrate semiconducting properties. For these materials, the *4f^6^* states of the Sm^2+^ and Eu *4f^7^* states lie between the valence band (formed by the *3p* orbitals of S) and the CB (constituted of the *5d* and *6s* orbitals of the lanthanides). In the case of SmS, the gap between the *4f* state and the bottom of the CB collapses at a pressure of about 0.65 GPa, providing metallic properties, or a low resistance state. 

Based on previous investigation on bulk crystals [6], it can be concluded that a mixture between the layers leads to a shift of the pressure threshold value of resistance drop. This could be related either to the size effects or the electronic structure, with the occurrence of a Eu *4f^7^* level deep in the energy gap that is less important in the SMT of the mixed system [33]. Eu is among the rare-earth elements which do not induce the valence transition in SmS, so external pressure is needed to induce the SMT in those mixed systems. Initial studies have indicated that the size factor is the most significant reason for the SMT, especially for substituent elements with smaller ion sizes than Sm^2+^. In that case, the bottom of the *5d* band lowers in energy, due to the local compression by the crystal lattice of the Sm ions. Nevertheless, other elements, such as Ca and Yb, though they show a smaller ion radius than Sm^2+^, do not manage to induce the SMT under ambient conditions. This means that the ionic radius is not the only factor determining the pressure threshold, but also the electronic structure plays a significant role. In our case of Eu-substituted SmS, the Eu ions are all divalent, as the trivalent charge state is not stabilized [34]. In contrast, for the semiconducting SmS, there is a mixture of Sm^2+^ and Sm^3+^ ions in the thin films, as witnessed from the derived lattice constant and XPS analysis. Hence the introduction of Eu^2+^ in a SmS thin film forces the (alloyed) SmS to acquire a slightly larger lattice constant [35]. Indeed, in case of the 35/30/35 nm SmS/EuS/SmS tri-layer deposited at 400 °C, the (200) peak position is located at about 30.2°, slightly lower than the corresponding value (~30.4°) in a 100 nm SmS film. Consequently, the larger lattice constant leads to a larger energy spacing for the *4f*-*5d* gap [6]. The latter is likely the main physical reason for the second drop at higher force values (~1.5 N). Increasing the amount of Eu, we notice the change from a discontinuous to a continuous transition (see Figure 8), which is in accordance with previous results [6]. As a matter of fact, the three states system presented in Figure 6a is a consequence of the occurrence of more than one material system in the entire thin film stack.

As an increase in temperature can enhance the interdiffusion of SmS and EuS, the tri-layer thin film shows a better mixing upon deposition at 400 °C, with Eu more equally spread out towards the outer parts of the stack (Figure 7b). Additionally, the outer regions show a more stoichiometric composition (in terms of the relative concentration of Sm and S) in comparison with the deposition at 250 °C. Also, prolonged post-annealing of 4 h at 400 °C (Figure 7c) did not show any reliable progress on mixing. Besides, the oxygen relative concentration increases only close to the material´s surface. Deposition at considerably higher substrate temperature than 400 °C would be needed for further investigate the mixing in the SmS/EuS/SmS system. Another option would also be the post-annealing at temperatures higher than 400 °C. Nevertheless, any attempt to provide further mixing should seriously take into consideration the corresponding HTXRD results (Figure 4a), which indicate the oxidation of the system at elevated temperature. 

Figure 7d,e show representative Sm 3d_5/2_ and Eu3d_5/2_ photoelectron spectra. The different traces correspond to measurements after 25 s (red curve), 150 s (navy curve) and 275 s (green curve) of argon ion sputtering. The XPS depth profiling process and the high surface reactivity of SmS can drastically change the ratio of the measured valence states of the lanthanide ions in the thin films [32]. Nevertheless, for the calculations of the element concentrations throughout the samples thickness, this is not an issue. For a further and detailed discussion of the valence state evaluation of Sm ions, as well as the surface oxidation of SmS thin films and the impact of depth profiling, we refer to the work done in [32]. 

Obviously, there is an intricate relation between the diffusion process (Figure 7) and the electrical changes (Figure 6) in the studied devices. For instance, in an inhomogeneous strongly correlated system, consisting of a semiconducting (EuS-like) and two metallic (M-SmS-like) phases, the measured property (e.g., resistance change) is an average value related to the entire system [36]. Also, the higher temperature deposition, at 400 °C, can provide an electronic state rearrangement, triggered by temperature [37]. This rearrangement can boost electrons either from the *4f* states of Sm or impurity levels within the energy gap to the CB. The measured electrical resistance, in the semiconducting state, can thus be lower upon deposition at 400 °C compared to films deposited at lower temperature, in accordance with our results (Figure 6a,c). Future work will focus on studying the influence of (measurement) temperature on the electrical behavior of these alloyed thin films, as this will yield insight in the dynamics of the metal-insulator transition [38].

In order to overcome the limited diffusion lengths, tri-layers consisting of thinner SmS and EuS layers were also deposited. Substrate temperature was chosen at 400 °C, in order to maximize the interdiffusion. Based on literature about Eu-doped SmS bulk crystals, we attempted to deposit two types of compositions, thus aiming at a different piezoresistive behavior. For a fully mixed tri-layer system that would show a discontinuous resistance change, the Eu concentration should be below the critical value of about 25% substitution [6]. For a second tri-layer, a higher Eu amount was chosen, in order to obtain a continuous resistance change. This is because only at lower Eu concentration is the resistance change due to the first-order valence change in the Sm ions. 

For the first type, we deposited a tri-layer SmS/EuS/SmS with thicknesses of 18/4/18 nm, yielding an overall Eu fraction of 10%. Despite the small thickness, the thin film is well crystallized with both (111) and (200) reflections prominently visible (Figure 8a). For the corresponding piezoresistance behavior (Figure 8b), only one drop is observed. This is probably a result of the small thickness of the intermediate EuS layer, which is now more homogeneously diffused into the outer SmS layers. In the case of using 50% of EuS (6/12/6 nm SmS/EuS/SmS), only the (200) XRD peak appeared (Figure 8a), while the resistance response to the applied force is almost perfectly continuous (Figure 8c). Nevertheless, around 1.1 N, a sudden drop in the resistance can be observed (Figure 8c), which points to the mixing not being fully completed. In both cases there is a change of roughly one order of magnitude in the resistance, when increasing the applied force from about 0 N to 1.5 N. Taking into consideration that we used the same top electrode material (Ir) with the same area as in the devices in Figure 6, the moderate resistance change should be attributed to either the lower structural quality, the smaller thickness or the occurrence of EuS-rich films, which show a rather limited change in resistance.

To shed light on the diffusion evolution, we used XPS depth profiling (with a sputter time of 3 s per step) for the case of 6/12/6 nm SmS/EuS/SmS (Figure 9). The two compounds homogeneously interdiffuse throughout the entire volume of the deposited tri-layer system, although some local variations could still occur. These results are in line with the previous analysis related to the resistive behavior.

## 4. Conclusions

In this manuscript, we reported on high-quality SmS/EuS/SmS tri-layer thin films deposited by e-beam evaporation. The structural properties were determined, as well as measurements of their resistance response to the applied force. This work demonstrates a well-defined hysteretic pressure-triggered semiconductor to metal transition (SMT) in SmS/EuS/SmS thin films. Depending on the substrate temperature and thus the degree of interdiffusion, we were able to demonstrate either a three or a two state piezoresistive system. Three state piezoresistive systems are not common and can for instance be used in nano-sensors, where they can selectively operate at different regimes of force, providing either a continuous or discontinuous change of electrical properties. In case of a 35/30/35 nm stack deposited at a substrate temperature of 400 °C, the resistance change tends to become continuous. A post-annealing at 400 °C up to 4 h did not lead to significant additional diffusion between the layers. HTXRD results on the nanocomposite system demonstrated that an oxidation process begins at 500 °C. Thinner nanocomposite layers were also deposited to evaluate the influence of thickness to the diffusion, showing improved mixing between the layers. This resulted in a more continuous piezoresistive response. The change in resistance for the studied pressure range is more limited as in the case of pure SmS thin films, since no semiconductor to metal transition occurs. This work shows promising experimental results on the piezoresistance response of SmS/EuS/SmS tri-layer nanocomposites. On the one hand, this triggers the further exploration of this system, where the degree of interdiffusion could be further controlled to arrive at specific piezoresistive responses. On the other hand, these developments support the future application in contemporary integrated piezo-based electronic devices, such as piezo-electronic memories and RF switches. Last but not least, studying the behavior of SmS thin films, using other substituting lanthanides, like Gd or Y, could also be future avenues.

## Figures and Tables

**Figure 1 nanomaterials-09-01513-f001:**
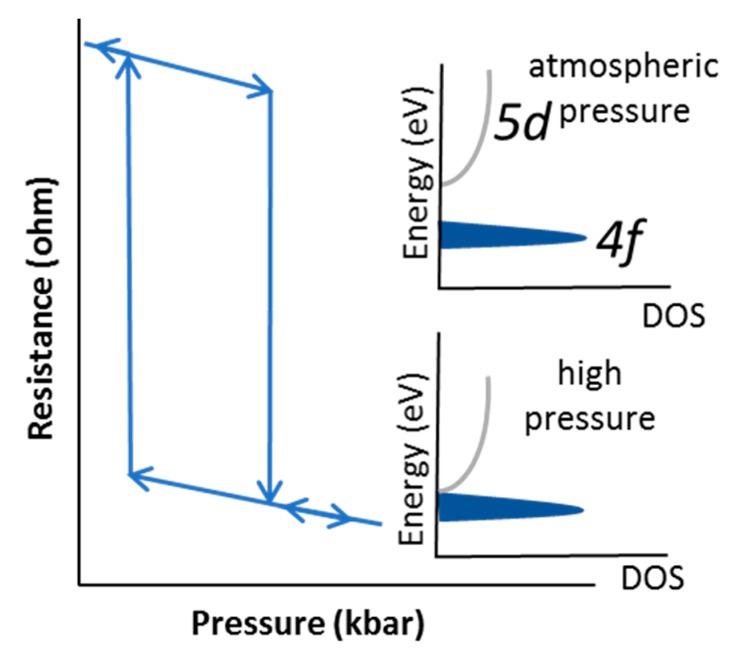
Schematic representation of the piezoresistive response of single crystal SmS. At the critical pressure, a change from the high-resistive to the low-resistive state occurs, due to the pressure induced shift of the *5d* conduction band towards the *4f* states of Sm^2+^ (inset). The top inset shows the gap between the *4f* states and *5d* conduction band at atmospheric pressure. The bottom inset demonstrates the closing of the gap, upon application of pressure. Upon release of the pressure, the SmS returns to the high-resistive state.

**Figure 2 nanomaterials-09-01513-f002:**
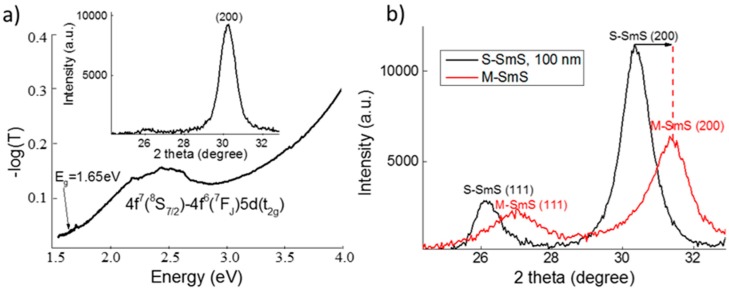
Basic properties of the deposited SmS and EuS thin films. (**a**) Optical (on glass) and structural (inset, on Si wafer) fingerprint of a 25 nm as-deposited EuS. (**b**) XRD patterns showing the structural behavior in the as-deposited (black line) S-SmS, as well as in the metallic state (red line), after rubbing. The lattice planes are indicated. In both thin film depositions, the substrate temperature was 250 °C.

**Figure 3 nanomaterials-09-01513-f003:**
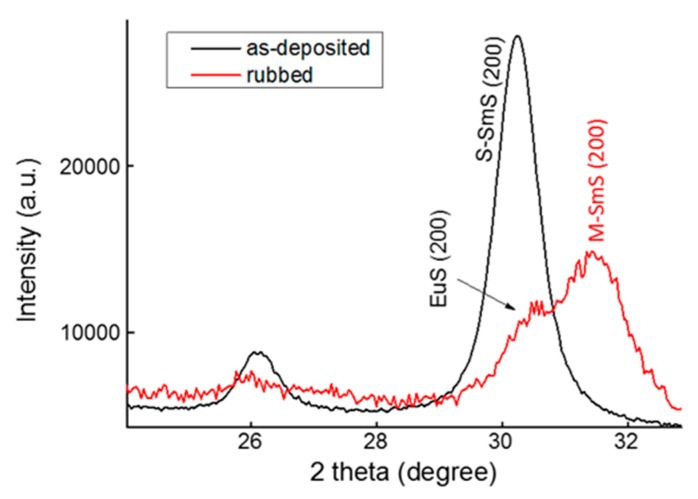
XRD patterns of an as-deposited SmS/EuS/SmS (35/30/35 nm) tri-layer deposited at 250 °C (black curve), and after rubbing (red curve).

**Figure 4 nanomaterials-09-01513-f004:**
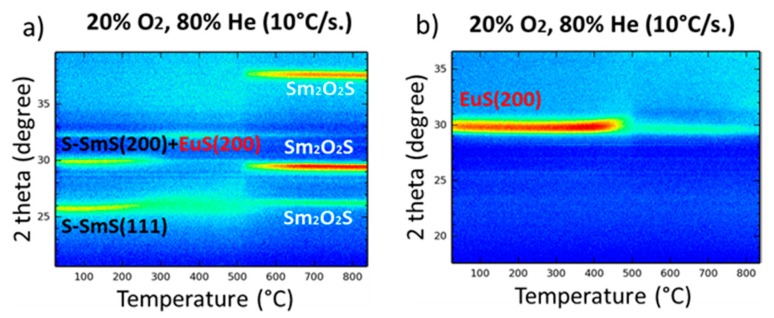
(**a**) In situ XRD patterns for increasing temperature of an as-deposited 35/30/35 nm SmS/EuS/SmS tri-layer in an ambient-like atmosphere of 20% O_2_ and 80% He. (**b**) Same as in (**a**) for an as-deposited 25 nm EuS thin film on Si. Both thin films were deposited at 250 °C.

**Figure 5 nanomaterials-09-01513-f005:**
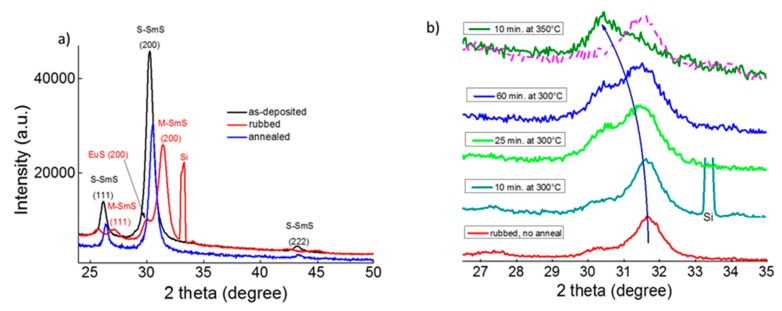
(**a**) XRD of an as-deposited SmS/EuS/SmS (35/30/35 nm) tri-layer at 400 °C (black curve), after rubbing (red curve) and after annealing in vacuum at 400 °C, for 10 min, and then cooling down (blue curve). (**b**) Cumulative annealing in ambient air of a similarly rubbed sample, as in (**a**), to provide the thermally triggered metal to semiconductor transition. For comparison, XRD pattern of a 100 nm SmS thin film, after 30 min annealing at 350 °C, is also depicted with the purple dashed line.

**Figure 6 nanomaterials-09-01513-f006:**
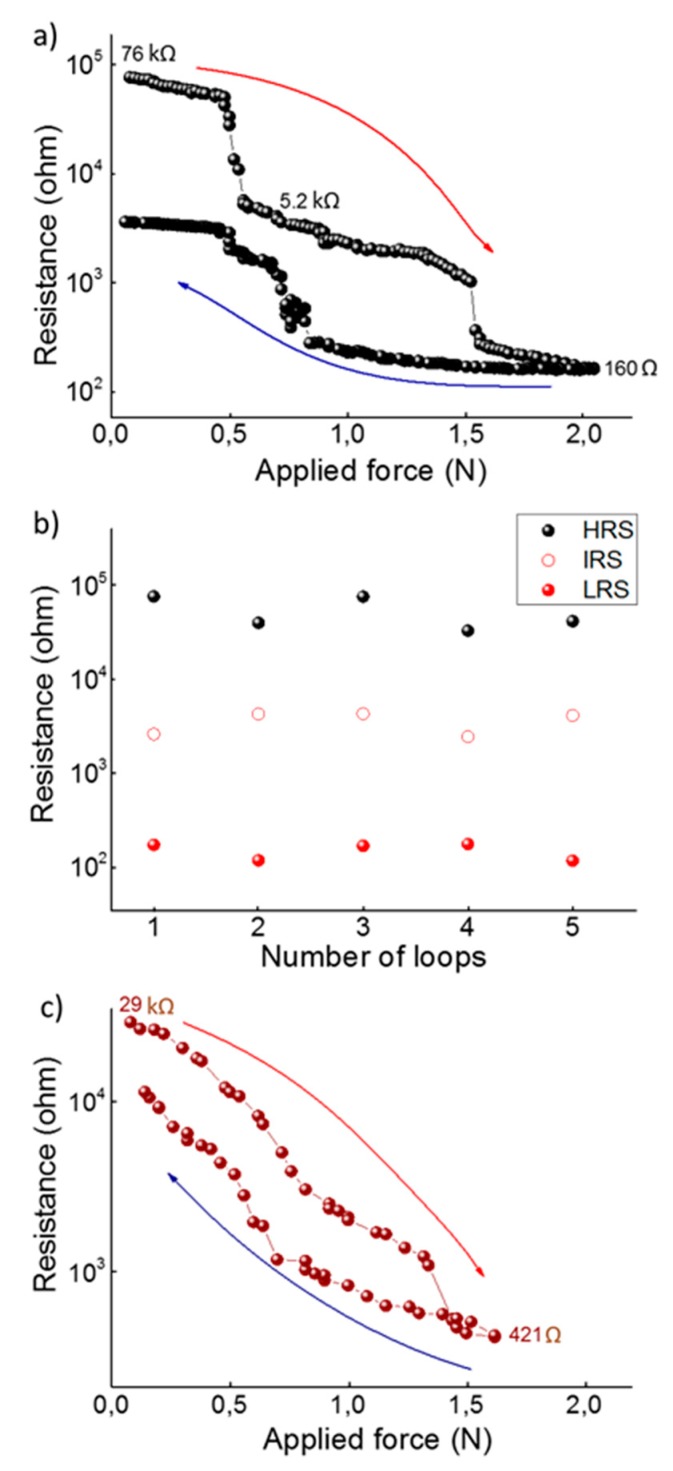
(**a**) Resistance across the SmS/EuS/SmS (35/30/35 nm, deposited at 250 °C) tri-layer, upon applying force up to 2 N. Red arrow for increasing force, the navy arrow when unloading. (**b**) Resistance value for five consecutive cycles of loading and unloading, at a force of 0.05, 0.9 and 1.6 N (corresponding to the high (HRS), intermediate (IRS) and low (LRS) resistance state) during the loading phase of the cycles. (**c**) Resistance of a thin film tri-layer with the same composition, deposited at 400 °C.

**Figure 7 nanomaterials-09-01513-f007:**
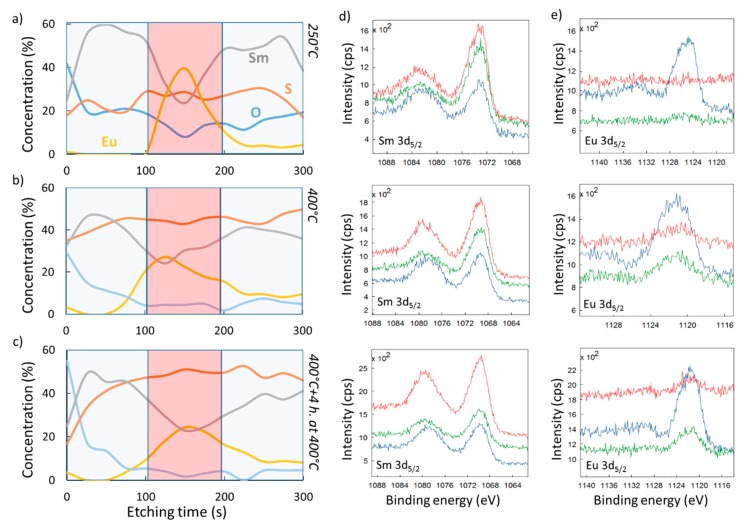
XPS depth profiling analysis of an as-deposited 35/30/35 nm SmS/EuS/SmS tri-layer system, deposited 250 °C (**a**), deposited at 400 °C (**b**), and after 4 h post-deposition annealing at 400 °C (**c**). (**d**) Sm 3d_5/2_ photoelectron peaks for the fabricated samples in Figure 7a,c, from top to bottom, respectively. (**e**) Eu 3d_5/2_ photoelectron peaks, as in (**d**). In both (**d**) and (**e**), the red curve corresponds to the photoelectron spectra after 25 s of sputtering, the navy one to 150 s, and the green curve to the sputtering time of 275 s.

**Figure 8 nanomaterials-09-01513-f008:**
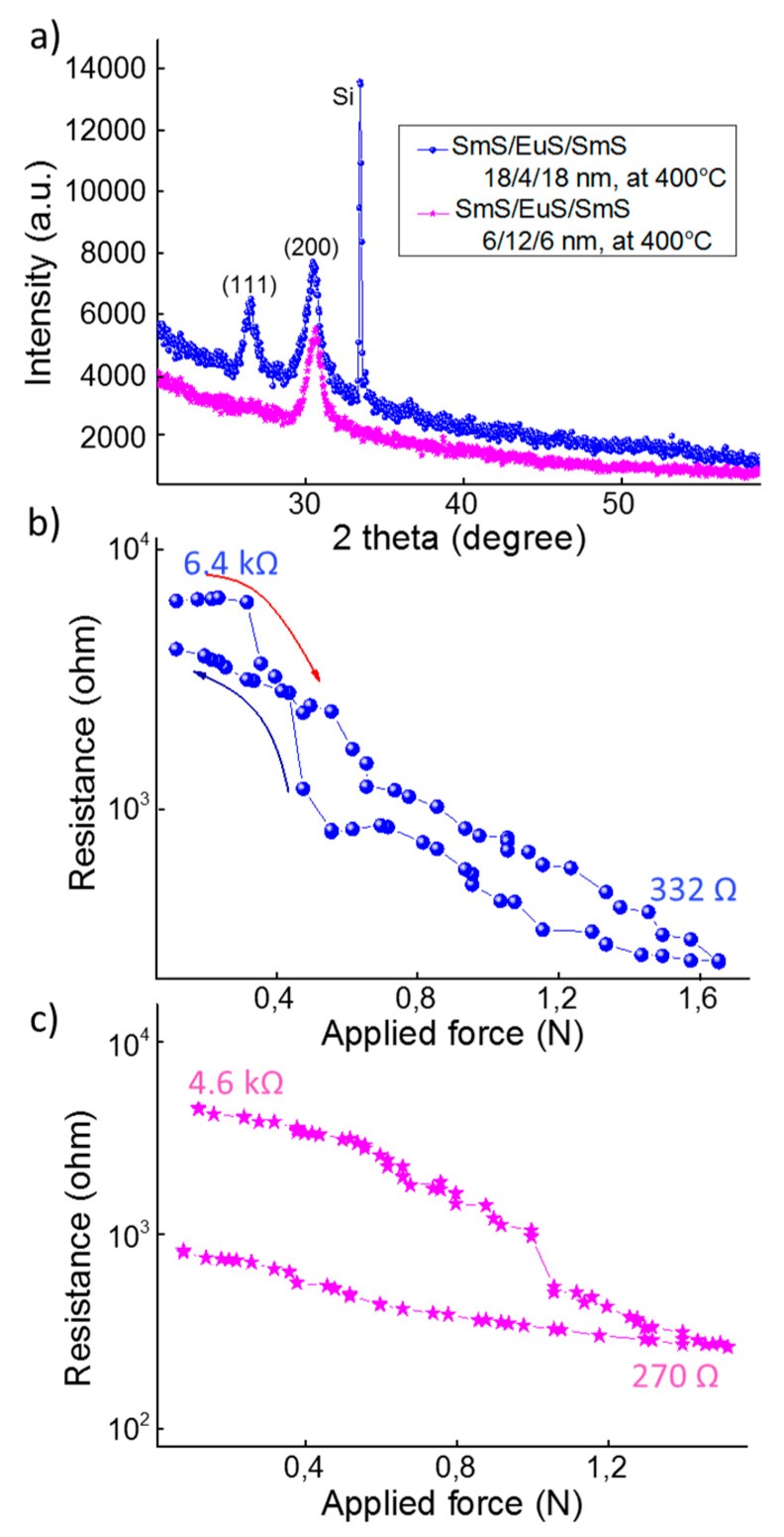
(**a**) XRD patterns of the thinner tri-layer systems (18/4/18 nm in blue dots and 6/12/6 nm in purple stars). Piezoresistance behavior for 18/4/18 nm tri-layer (**b**) and 6/12/6 nm tri-layer (**c**) deposited in between Ir electrodes. For clarity, in (**b**) red arrow represents the loading process, while the navy arrow shows the unloading process.

**Figure 9 nanomaterials-09-01513-f009:**
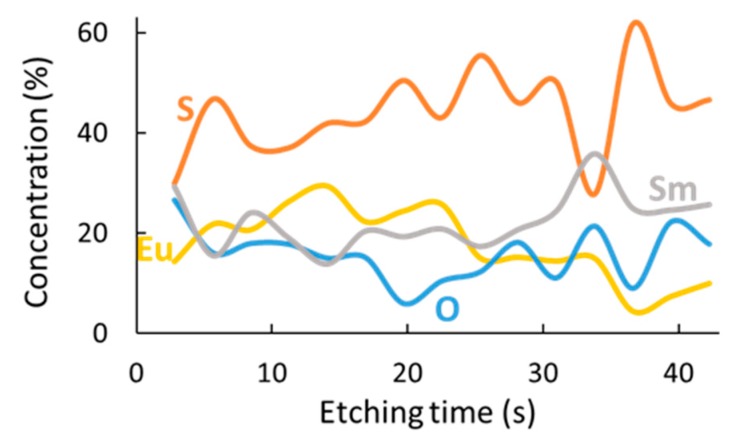
XPS depth profiling for the 6/12/6 nm SmS/EuS/SmS tri-layer system deposited at 400 °C.

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
