# Peer review of "SmS/EuS/SmS Tri-Layer Thin Films: The Role of Diffusion in the Pressure Triggered Semiconductor-Metal Transition"

_nanomaterials, 2019, doi:10.3390/nano9111513_

Round 1

Reviewer 1 Report

The goal of this report is to investigate pressure-induced  metal-semiconductor transitions in tri-layer SmS/EuS/SmS films. The hysteretic resistance loops behaviour was investigated for SmS at different thermal treatment of the layer. The diffusion effect between the proper layers is investigated by the XPS depth profiling analysis. The Authors demonstrate three or two state piezoresistive  system depending on the diffusion intensity. The work is interesting, the asset of the work can be any applications in the construction of new devices. On the base of XRD and XPS analysis two different valence states of Sm ions in the SmS films are reported, however, I did not find the respective XPS spectra, this could be interesting for many researchers. Despite this, the work is still suitable for publication as it is. I recommend this manuscript to be published in Nanomaterials.

The work indicates new results on the piezoresistance response of SmS/EuS/SmS trilayer nanocomposites. Namely, the Authors documented hysteretic pressure-triggered semiconductor to metal transition in SmS thin films with accompanying change of the lattice parameter. Therefore, the x-ray diffraction analysis of the investigated films obtained at different heat treatment is reported. The Authors suggest the future application of investigated system the application in contemporary integrated piezo-based electronic devices.
The elements Eu and Sm may change their valence depending on the temperature or applied pressure. The valence of Sm was determined from XPS measurements. I suggest to show the respective core-level XPS spectra. The obtained results show how external conditions (T, P) change the electrical properties of the system due to diffusion process and valence change of Sm, in this respect the investigations are interesting.
The paper is well written. In my opinion, the work is a research in terms of application, it analyzes in detail the impact of external conditions on the electric transport of layers, and this topic could be interesting. I maintain my positive opinion, although the presentation of the core-level XPS spectra may be interesting for physicists. 

Reviewer 2 Report

This manuscript (MS) reports on possibility of application for the role of diffusion on the pressure triggered semiconductor metal transition in a device of SmS/EuS/Sms which is a representative material of the pressure-induced metal-insulator transition. In particular, a thermal annealing effect is focused for the generation of pressure effect. It is possible as a piezoresistance sensor. The idea using the diffusion effect is distinguished. The given explanations about several figures are reasonable. The insulator-metal transition of jump displayed by annealing of 250oC becomes broad at 400oC and the magnitude of the jump is reduced by annealing, as shown Fig.s 6 (a) and (c). If these phenomena are explained by using theories of the insulator-metal transition and Figure 7, this MS is largely enhanced in quality. I would like to recommend two papers; Physical Review Letters 118, 036602 (2017); New Journal Physics 6, 52 (2004). The former is theoretical, but I think the result can be studied, the latter can give a concept for explaining the diffusion effect; section 2 “A new method to observe the Mott transition” of the second paper can help you. Except for my comments, I could not find errors.

Round 2

Reviewer 2 Report

I think this manuscript was highly enhanced in quality.